# Toxic Indoor Air Is a Potential Risk of Causing Immuno Suppression and Morbidity—A Pilot Study

**DOI:** 10.3390/jof8020104

**Published:** 2022-01-21

**Authors:** Kirsi Vaali, Marja Tuomela, Marika Mannerström, Tuula Heinonen, Tamara Tuuminen

**Affiliations:** 1SelexLab Oy, Kalevankatu 17 A, 00100 Helsinki, Finland; 2Co-op Bionautit, Viikinkaari 9, 00790 Helsinki, Finland; marja.tuomela@helsinki.fi; 3Department of Microbiology, University of Helsinki, 00014 Helsinki, Finland; 4The Finnish Centre for Alternative Methods, Faculty of Medicine and Health Technology, Tampere University, Arvo Ylpön katu 34, 33014 Tampere, Finland; marika.mannerstrom@tuni.fi (M.M.); tuula.heinonen@tuni.fi (T.H.); 5Medical Center Kruunuhaka Oy, Kaisaniemenkatu 8B a, 00100 Helsinki, Finland; tuuminen@gmail.com

**Keywords:** indoor air water, dampness and mold hypersensitivity syndrome, mycotoxins, clinical toxicology, sick building syndrome, urine

## Abstract

We aimed to establish an etiology-based connection between the symptoms experienced by the occupants of a workplace and the presence in the building of toxic dampness microbiota. The occupants (5/6) underwent a medical examination and urine samples (2/6) were analyzed by LC-MS/MS for mycotoxins at two time-points. The magnitude of inhaled water was estimated. Building-derived bacteria and fungi were identified and assessed for toxicity. Separate cytotoxicity tests using human THP-1 macrophages were performed from the office’s indoor air water condensates. Office-derived indoor water samples (*n* = 4/4) were toxic to human THP-1 macrophages. *Penicillium*, *Acremonium sensu lato*, *Aspergillus ochraceus* group and *Aspergillus* section *Aspergillus* grew from the building material samples. These colonies were toxic in boar sperm tests (*n* = 11/32); four were toxic to BHK-21 cells. Mycophenolic acid, which is a potential immunosuppressant, was detected in the initial and follow-up urine samples of (2/2) office workers who did not take immunosuppressive drugs. Their urinary mycotoxin profiles differed from household and unrelated controls. Our study suggests that the presence of mycotoxins in indoor air is linked to the morbidity of the occupants. The cytotoxicity test of the indoor air condensate is a promising tool for risk assessment in moisture-damaged buildings.

## 1. Introduction

There is convincing scientific data that the exposure to dampness microbiota and the decay products of construction materials may cause the development of polymorbidities and systemic inflammation [1,2], now collectively called dampness and mold hypersensitivity syndrome (DMHS) [3], although this designation has not yet received official recognition. The causality between the exposure to dampness microbiota (the source) and the clinical outcome (the effect) is still being questioned. Polymorbidity in the occupants of moisture-damaged buildings with indoor air toxicity has been extensively studied in Finnish schoolchildren and the occupants of a moisture-damaged hospital and a police station [1,4,5,6,7,8]. These studies have demonstrated that the prolonged exposure to dampness microbiota and the decay products of a building’s construction materials may cause a plethora of non-respiratory symptoms such as profound fatigue, neurological, gastrointestinal, and muscular-skeletal problems in addition to the already acknowledged respiratory symptoms. Because no single biomarker to diagnose DMHS has yet been devised, the diagnosis remains clinical and is based on interview. The patients often recall some form of water leakage in their homes or workplaces and a gradual worsening of their symptoms. Initially, the symptoms are reversible and related to the so-called sick building syndrome, (SBS) [9]. However, if there is continuous or cumulative exposure (e.g., starting already in the kindergarten, and then in school, high school and then workplaces) the symptoms may become aggravated and progress into irreversible multi-organ DMHS [3]. Importantly, DMHS is associated with an increased rate of respiratory infections [3] probably indicating decreased immunologic tolerance to environmental viruses and bacteria. Moreover, deprivation of the immune system has been proposed by us when reviewing the long-term morbidity among the occupants of one of the moisture-damaged school [10]. A higher prevalence of different oncological diseases, unexplained succeptibility to sepsis and a higher than normal rate of autoimmune diseases have been reported [10].

The investigations of the indoor air quality do not always recognize existing problems within buildings; for example, the impact of air toxicity has been overlooked. Most of the toxic metabolites produced by fungi or bacteria recovered from moisture-damaged buildings have a molecular weight of 300–2000 g/mol, i.e., they are non-volatile. However, mycotoxins may be present in indoor air either attached to fungal spores or to fragments of these microorganisms, e.g., broken or fractured conidia and hyphae. These particles may remain suspended in the air for a long time [11]. Recently, it was reported [12,13,14] that some species of dampness microbiota produce vesicles, or the so-called “guttation droplets”, or exudates on a culture dish. These droplets may contain substances which are toxic to eukaryotic cell types at dilutions of 1/100–20,000 [12,15,16]. The most efficient solvents for mycotoxins are relatively polar solvents, such as methanol or ethanol, only a few mycotoxins are water soluble [17]. The relative humidity (RH %) of the air becomes elevated when the environment is crowded with occupants (e.g., schools during the working hours) which allows these droplets to move around aerodynamically and thus the toxins can spread throughout the environment [18]. For example, *Penicillium expansum* recovered from gypsum boards produced these kinds of toxic droplets, i.e., exudates that were released into the air and were >100-times more toxic in the cell culture assays than indoor air isolates of *Aspergillus, Chaetomium, Stachybotrys* and *Paecilomyces* [13]. These droplets may represent an even greater health hazard than the spores and nanoparticles of a microbial mass growing in the environment.

To address the issue of air toxicity, we have devised a novel method of collection of indoor air water vapour by adopting a newly patented technique with the so-called E-collector: (US Patent 10,502,722 B2; www.sisailmatutkimuspalvelut.fi, accessed on 21 November 2021) [18,19]. Briefly, the air condensate is collected on a cooled surface of a steel plate, the condensate then can be tested for its cytotoxicity against living cells [19]. Boar sperm cells [20,21] or several eukaryotic cell lines, such as porcine kidney cell line (PK-15) [13,22,23] or murine neuroblastoma cells (MNA) [13,23,24], have been used in toxicity studies instead of experiments in production animals. In this publication, we have successfully used boar spermatozoids and baby hamster kidney (BHK) cells for the same purpose.

Here, we describe a plausible pathway of how the mycotoxins present in the indoor air water samples gain access to the exposed individuals. We demonstrate the usefulness of urinalysis for the detection of mycotoxins as biomarkers of DMHS. A novel method [19] for the sampling and testing of indoor air toxicity is presented and shortly discussed.

## 2. Materials and Methods

### 2.1. The Patients

This working community comprised six members, and their work premises which were located partly below ground level in an office building (Figure 1). Within approximately three months after the move into this office, every member of this workplace community started to suffer from symptoms compatible with their exposure to indoor air dampness microbiota. Finally, the symptoms became unbearable. The patients wanted to remain anonymous and therefore their gender and age are not reported.

Five (Patients 1–5) out of six occupants of the problematic building were examined by one of us (TT) and one (Patient 6) was interviewed over the telephone. All were healthy before the move into the problematic office. Their life histories and the symptoms were recorded during their visits and then retrieved with the patients’ written consent.

For comparison of urinalysis profiles, we investigated two unrelated patients (Patient X and Y) who volunteered to have their urine samples tested because their symptoms were compatible with an exposure to dampness microbiota. They were unrelated to each other, they lived and worked in different cities than the working community in question. Neither microbiological nor toxicological investigations in their homes or workplaces had been performed at the time of our investigation. Patient X moved with all his/her belongings to a new home but continued to experience a general feeling of sickness, fatigue, and mucosal irritation.

### 2.2. Detection of Mycotoxins in the Patients’ Urine Samples

The detection of mycotoxins from the patients’ urine was performed in the Great Plains Laboratory Inc. (Lenexa, KS, USA) using a liquid chromatography tandem mass spectrometry (LC-MS/MS) technique. The urine specimens from Patients 1 and 2 of the working community and a sample from a household control (HHC) of Patient 1, and from Patients X and Y were analyzed. Follow-up samples from Patients 1 and 2 were taken 2.5 months later after the working community had moved to a clean building.

The detection of mycotoxins was performed according to the standard operational procedure of the laboratory. The presence of the mycotoxin mycophenolic acid was confirmed after extraction of urine and LC/MS/MS testing of the extract with a deuterated mycophenolic acid internal standard [25], indicating that a peak with all the characteristic ions of authentic mycophenolic acid [26,27] and the identical retention time of mycophenolic acid was present in many patients [28]. In addition, patients with high mycophenolic acid in urine has been tested as positive for mycophenolic acid by a commercial specific immunoassay (Cedia) from ThermoScientific for mycophenolic acid in urine. Mycophenolic acid can be produced by multiple species of Penicillium and other molds [29,30,31].

### 2.3. The Office Building

The working community had their office located on the ground floor of a building with a cellar. It was built from concrete elements and had a mechanical ventilation system. One of the walls in the office faced a non-heated hollow area which was below the soil surface and was lined on the bottom with a layer of gravel and on the top with a closing plate mostly without waterproofing. Thus, water from rain and melting snow had leaked inside the hollow area enabling microbial growth. There were several air leaks in the wall of the office adjacent to the hollow area. Because of the negative pressure difference (−12 Pa on average, momentarily even −40 Pa) in the office compared with the hollow area, the air from this area was able to penetrate the office. The hollow area remained moist all year round.

The office was also examined by sniffer dogs which are trained in mold detection. The dogs marked six locations, with five of these being beside the wall facing the hollow area. In some locations, the characteristic *o**dors* of *microbes* were noticeable also to the occupants.

### 2.4. Sampling of the Indoor Air Condensate Water

The device and the technique to collect indoor air water samples have been described elsewhere [19]. Briefly, the principle of the collection is based on the phenomenon of the condensation of water molecules on the top of cold surfaces of metal plates. After melting of the frozen water to room temperature, the condensate is collected from the tray of the device into Eppendorf tubes that are transferred to the laboratory where the condensate is analyzed in a cell culture assay. This collection technique enables the harvesting of airborne toxic substances in indoor air vapor that may contain mycotoxins [13] as well as various large molecules [32].

The condensing water samples from the six locations marked by the sniffer dogs were collected with the E-collector from the office building. Samples were also collected from the home of Patient 1.

### 2.5. Indoor Air Toxicity Studies

The toxicity of the indoor air water condensates collected from different locations of the office and from the home of Patient 1 was studied using the human acute monocytic leukemia cell line (THP-1) using water-soluble tetrazolium salts (WST-1) in a cytotoxicity assay [19,33]. The WST-1 assay is an indicator of mitochondrial activity and commonly used to assess cell viability; decreased mitochondrial activity reflects a loss of cell viability, i.e., cell death. Increased mitochondrial activity reflects an increased cell number, i.e., proliferation, or increased cellular respiration induced by mitochondrial uncoupling reactions [34,35]. The WST-1 Cell Proliferation Reagent was obtained from Roche (Basel, Switzerland). Human THP-1 monocytes (Cat. No. TIB-202) were from ATCC (LGC Promochem AB, Boras, Sweden), and were verified to be *Mycoplasma*-free (MycoAlert™ kit, Lonza Basel, Switzerland) prior to use. THP-1 monocytes were maintained in RPMI 1640 Medium supplemented with 10% fetal bovine serum (FBS), and when differentiated to macrophages, challenged with 25 nM phorbol 12-myristate 13-acetate (PMA) (Sigma Aldrich, Steinheim, Germany) for 48 h followed by a 24 h recovery period.

The cells were seeded into 96-well plates at a density of 10^5^ cells/well and exposed for 24 h to the indoor air condensates at 10% and 25% concentrations (in RPMI supplemented with 5% FBS). Cells exposed to an equal volume of distilled water (10% or 25%) served as negative controls, and to nickel II sulphate hexahydrate (2.0 and 20.0 µg/mL) as a positive control. All samples and controls were tested in six replicates. In the assessment of cell viability, 10 µL/well WST-1 reagent was added to the cells for 2 hours, and subsequently absorbance was read at 450 nm. The absorbance is directly proportional to the mitochondrial activity (cell viability). The absorbances were normalized, i.e., the untreated control was set as 100%, and all other data were calculated relative to the control absorbance as either % decrease in cell viability (negative values) or % increase in proliferation (positive values). The statistical significance of the changes compared to the untreated control was tested with Student’s *t*-test.

The indoor air condensates from the office were tested using both THP-1 monocytes and THP-1 macrophages at a 10% sample concentration. The condensates from the home of Patient 1 were tested at two concentrations, i.e., 10% and 25%, but using only THP-1 macrophages.

### 2.6. Detection of Mycotoxins from the Condensed Indoor Air Sample

A left-over aliquot of 0.5 mL of the condensed indoor air water sample in the Eppendorf tube was sent at ambient temperature for LC-MS/MS analysis to the Great Plains laboratory. This sample had not been stored protected from light.

### 2.7. Indoor Air Relative Humidity Measurements and Estimation of Respiratory Mycotoxin Exposure

The relative humidity was measured with Gann Hydromette BL RH-T (Gann Mess- u. Regeltechnik GmbH, Gerlingen, Germany). To show the probability of the inhalation exposure route instead of the more improbable oral exposure, we estimated the amount of water inhaled in the prevailing indoor air conditions (Table 1). The estimation is approximately 14–15 g (mL) of inhaled water/day, resulting in a challenge of 730–760 g of water/2.5 months exposure time.

The estimated volume of inhaled air by adults is 0.5 L/inhalation, the frequency of inhalations was estimated to be 13 times/min (the average 12–16 times/min, depending on the physical activity). The estimated frequency of inhalations is 13/h in office workers. The humid ratio was calculated from the measured relative humidity (RH) and room temperature (°C), according to the formula: available online: http://www.flycarpet.net/en/PsyOnline (accessed on 21 November 2021).

### 2.8. Collection of a Sample for Microbiological Analysis

A gravel sample was collected from the hollow area in the proximity of the floor of the office. There was no visible fungal growth in the sample. The sample was cultured on three agar media, namely malt extract (MEA) with the following formulations: malt extract, 20 g; agar, 15 g; and distilled water to 1 liter [36], dichloran-glycerol (DG18) with the following formulation: glucose, 10 g; bacteriological peptone, 5 g; KH_2_PO_4_, 1.0 g; MgSO_4_ × 7H_2_O, 0.5 g; agar, 15 g; distilled water to 1 / L; 220 g of glycerol (analytical reagent grade) having a final concentration of 18% (wt/wt); and 1 mL of a 0.2% (in ethanol) solution of dichloran (2,6-dichloro-4-nitroaniline) to a final concentration of 2 mg / L [36] and tryptone glucose yeast extract (TGY) with the following formulation: tryptone, 5.0 g; yeast extract, 2.5 g; glucose, 1.0 g; agar 15 g; and distilled water to 1 L [37]. All the media were autoclaved for 15 min at 121°C, and pH values for ready medium were 5.5, 5.6 and 7.0, respectively.

Two inoculation methods with two replicates were used for each medium and dilution: (1) Direct inoculation on the agar plates (0.5 g/plate). (2) Serially diluted suspensions from the sample were pipetted onto agar plates (0.1 mL/plate) according to the instructions of National Supervisory Authority for Welfare and Health in Finland [38,39]. The composition of the diluent is as follows: 0.0425 g KH_2_PO_4_, 0.25 g MgSO_4_ × 7 H_2_O, 0.008 g NaOH /1 L deionized water. The diluent was prepared as follows: adjusted to pH 7.0 ± 0.2; add 0.2 mL Tween 80 as the detergent, then autoclaved at 121 °C, 15 min. The plates were incubated at room temperature and the cultures were examined after one, two or three weeks with the fungal genera being identified by microscopy [40,41,42,43], and furthermore *Aspergillus* spp. was identified at a species or section level [44]. The fungal identification was based on the inspection of the morphology as recommended by the Finnish authorized bodies, although the current nomenclature has been recently changed in accordance with the advances in molecular methods [38]. The bacterial colonies were counted and *Streptomycetes* were identified [38].

### 2.9. Detection of Toxicity from the Microbial Growth

The colonies of all the fungal and bacterial species (*n* = 32) that grew on the agar plates were tested in the boar sperm test according to Andersson et al. [20] and Castagnoli et al. [21], and in a mammalian cell test according to Rasimus et al. [45]. A colony or a part of it (20–30 mg) was extracted into 96 % ethanol (0.2 mL). In the toxicity tests, boar spermatozoa that are used in artificial insemination (Figen Oy, Seinäjoki, Finland) and baby hamster kidney cells (BHK-21 [C13] ATCC^®^ CCL10™) were exposed to the ethanol extracts (*n* = 32). After incubation with the ethanol extracts, the inhibition of BHK-21 cell growth was assessed using the resazurin reduction assay [46], while toxicity to boar spermatozoa was assessed as a lack of motility observed with the CellSens computer program [21] and visually with a microscope [20]. Pure ethanol was a negative control in both tests. The BHK cells were maintained in Dulbecco’s modified Eagle’s medium (DMEM) supplemented with 10% fetal bovine serum (FBS), 2 mM L-glutamine, 100 U/mL penicillin and 100 µg/mL streptomycin.

### 2.10. Ethical Considerations

All participants of this study provided their written informed consent to use their records.

## 3. Results

### 3.1. Clinical Picture in Persons Exposed to Toxic Indoor Air

The symptoms reported by Patients 1–5 are summarized in Table 2. Patient 6 started to experience a very severe headache that did not become better during the weekends. This person was given a diagnosis of tension neck, and the symptoms were not interpreted as being associated with an exposure to poor indoor air quality. The patient was often on sick leave and changed the workplace before this investigation started. The work ability returned after changing the employer, and the headache did not reappear.

### 3.2. Detection of Mycotoxins from the Urine Samples

The mycotoxin profiles of Patients 1 and 2 and HHC, Patients X and Y, the controls, are shown in Table 3. It is noteworthy that the HHC for Patient 1 showed evidence of the excretion of ochratoxin A in the urine but the profile was different from that of Patient 1 and there was no mycophenolic acid detectable in his/her urine. The LC-MS/MS plots of mycophenolic acid are presented in Figure 2A,B. The mycotoxin profiles of Patients X and Y did not contain mycophenolic acid, and were different from those of Patients 1 and 2 who represented the work community exposure cohort.

### 3.3. Evaluation of the Office before Our Investigations

In addition to the notable negative pressure difference and air leaks (from −12 to −40 Pa), the assessment of the office revealed insufficient ventilation and a water damaged area in the ceiling covered with gypsum board. The cultured ceiling material contained 10^7^ colony forming units/gram (cfu/g) (on MEA plates) or 7 × 10^6^ cfu/g of fungi (on DG18 plates), and 3 × 10^7^ cfu/g of bacteria (on TGY plates). The identified fungi were *Aureobasidium* sp. (88–97%), and yeasts. It should be emphasized that these counts exceed the cut-off values, which are 10^4^ for fungi and 10^5^ for bacteria [38].

The sniffer dogs marked six locations, five of which were beside the wall facing the hollow area from where a sample was taken for microbiological analysis.

### 3.4. Detection of Mycotoxins from the Water Condensate

After the toxicity studies, a left-over sample from the condensed air was analyzed by the LC-MS/MS. No other peaks of mycotoxins were detected except for mycophenolic acid that was however below the limit of accurate detection. The absence of mycophenolic acid could not be ascertained beyond doubt from the indoor air sample because this sample had not been protected from light and this might have led to the degradation of mycophenolic acid.

### 3.5. Toxicity Studies from the Water Condensates

Toxicity studies from the office’s indoor air are shown in Table 4, those from the home of Patient 1 and the HHC are presented in Table 5. All indoor air condensates (tested at a 10% concentration in the final cell culture test system) collected from the office caused adverse effects on cells: Samples from 2 locations out of 4 induced THP-1 monocyte proliferation, whereas samples from all 4 locations were toxic to the THP-1 macrophages. None of the indoor air water condensates collected from the home of Patient 1 and HHC were toxic to THP-1 macrophages tested at the same i.e., 10% concentration. However, after increasing the indoor water condensate amount from 10% to 25% in the test system, toxicity was observed in the home of Patient 1 and the HHC. The samples taken from the bedroom located on the 1st floor, and that from the living room located on a 2nd floor induced a slight proliferation of THP-1 macrophages. Samples collected from the home were not tested on THP-1 monocytes, Table 5.

### 3.6. Microbiological Analysis and the Toxicity of the Cultured Microbial Colonies

The microbial growth from the office gravel sample was abundant on all the plates: the total fungal colonies were 13,000 cfu/g on MEA and 3900 cfu/g on DG18, and the total bacterial colony count was 42,000 cfu/g on TGY. *Penicillium* and *Acremonium sensu lato* constituted 97% of the fungal colonies on MEA plates, and on DG18 plates 92% of colonies were only *Acremonium sensu lato*. The other fungi, namely *Aspergillus* section *Aspergillus* (formerly *Eurotium*), *Aspergillus ochraceus* group, *Cladosporium* and *Trichoderma* grew only as a few colonies. No *Streptomyces* were detected.

Altogether, 11 microbial colonies (10 fungi and one bacterium) out of 32 colonies tested were toxic to the cultured cells (Table 6). All 11 colonies were toxic in the boar sperm test, and four of them were also toxic to the BHK-21 cells. The species that were toxic in both tests belonged to the fungal genera *Acremonium sensu lato*, which grew on MEA plates, and *Aspergillus* section *Aspergillus*, which grew on DG18 plates. The fungi which were toxic in the boar sperm test were only from the *Aspergillus ochraceus* group, which grew on DG18 plates, as well as from several *Penicillium* species, which grew on DG18 or MEA plates.

## 4. Discussion

This pilot study revealed the presence of mycotoxins in the urine of the occupants who had been exposed to toxic indoor air, and emphasizes that a urinalysis for mycotoxins, when combined with a careful patient history and medical check-up, is a valuable tool in the diagnosis of DMHS. Until today, exclusively gaseous and particle exposure of toxic indoor air has been investigated, but the condensed water component of the air has been largely overlooked [7,19]. Therefore, our condensed water approach followed by cytotoxicity assays provides a rationale of conducting a risk assessment producing a numerical outcome. Recently, a call for such health risk-based indoor air methods has been published [47].

We detected a high concentration of mycophenolic acid in the first and follow-up urine samples of the two occupants of the problematic office. Moreover, traces of mycophenolic acid in the condensed water sample were also evident. It is assumed that the content of mycophenolic acid would have been higher if the sample tubes been protected during their sampling, storage, transportation and analysis; unfortunately, this was not the case. The fungal species recovered from the office are a potential source of mycophenolic acid production. Although the amount of toxin might be small in terms of absolute values, the exposure is, however, significant because large volumes of indoor air are inhaled each day. This exposure is dependent on the relative humidity (RH %) of the air, i.e., when the RH % and temperature increases, the quantity of inhaled mycotoxins will increase. Toxins can also be inhaled along with fungal particles.

All occupants of this working community experienced symptoms that were compatible with the advanced SBS, or DMHS (Table 2). The mycotoxin profiles of the two occupants exposed to the same dampness microbiota were similar but were different from the profiles of two unrelated patients (Patients X and Y, Table 3). Patients X and Y also suspected that they had been exposed to dampness microbiota. They had also complained of symptoms, but the composition of dampness microbiota at the species level in their homes or workplaces must have been different. Therefore, these patients were taken as putative disease-controls for Patients 1 and 2.

Mycotoxins are secondary metabolites, and their production is highly species specific. Although mycotoxin production is common among indoor and outdoor fungi, some species are not thought to be mycotoxin producers, for example *Cladosporium* spp. [48]. Mycophenolic acid can be produced by the genera *Penicillium* [27] and *Aspergillus* section *Aspergillus* (formerly *Eurotium*) [49]. It was observed that several *Penicillium* species grew abundantly from the gravel sample taken from the hollow area in the proximity to the office; this was a location from which there was an air flow into the office. Four of the tested *Penicillium* species were found to be toxic in the boar sperm test as was the *Aspergillus* section *Aspergillus* which also grew from the gravel samples; this mold was not only toxic in the boar sperm test but also in the BHK-21 test. Half of the colonies of *Acremonium sensu lato*, which was a fungal group thriving in the gravel sample, was also shown to be toxic in both tests. There are only a few reports on the mycotoxins produced by *Acremonium* species. For example, it has been demonstrated that *Acremonium exuviarum* isolated from the building material produce a toxin called acrebol [24].

Previously, fungal identification was very difficult but now DNA-aided identification has proved to be beneficial. Here, we used conventional identification techniques established for the standard operational procedure of the accredited laboratory. It has been reported that a few fungal species may produce mycophenolic acid [50] and the species identified in this study belonged to the potential producers of mycophenolic acid. Importantly, we collected indoor air water condensate samples from other moisture-damaged offices, and six out of ten were also positive for mycophenolic acid (data not published). These samples were also toxic in the THP-1 tests. These findings suggest that mycophenolic acid is a common contaminant in Finnish moisture-damaged buildings.

The route of entry of mycotoxins into the body has been a topic of debate. For example, the urine of 3000 Swedish adolescents was recently analyzed with the presence of 35 different mycotoxins being identified [51]. It was speculated that children had been exposed by oral ingestion of contaminated foodstuffs. Instead, the exposure via indoor air was not considered although it is recognized that there are moisture-damaged buildings in Sweden and therefore the possibility of inhalation exposure should not have been excluded.

The oral intake of mycophenolate mofetil by Patients 1 and 2 was ruled out because they were not receiving any immunosuppressive medication. Taking into account the growth of potential mycophenolic acid producers in the office building, it does seem that the exposure occurred via inhalation of the office’s air supply. Mycophenolate mofetil is a first-line immunosuppressive drug used in transplantation immunology [52]. It is used to inhibit the proliferation of B and T lymphocytes and to prevent graft rejection and is also administered in the therapy of lupus nephritis [53]. If our pilot results are corroborated in a larger trial, we may be able to provide mechanisms by which some individuals exposed to dampness microbiota often experience recurrent infections [3]. These infections point to a dysfunction of the individual’s immune system. This can be either local through the inhibition of innate immunity of the cilia cells lining the mucosal epithelium, or systemic involving acquired immunity. Prolonged exposure to mycophenolic acid may further result in a dysregulation of immune checkpoints leading to uncontrolled cell proliferation e.g., cancer. We have already reported an epidemiological observation that a prolonged exposure to dampness microbiota is associated with higher oncological morbidity [10].

Another interesting finding emerging from this study was the excretion of citrinin by Patient 2. This toxin can be produced by *Acremonium* and *Penicillium* species [54]. Both genera grew abundantly from the gravel sample. Fungi of the *A. ochraceus* group are known to produce ochratoxin A (OTA) and this potent mycotoxin was detected in minor amounts in the urine of Patients 1 and 2. This fungus was present only as a few colonies on the agar plates.

OTA was found also in the urine of the HHC to Patient 1. This finding is consistent with the toxicological studies from the bedroom of both Patient 1 and the HHC (Table 5), however, no microbiological studies were undertaken in their home. The HHC worked from home in their apartment, whereas Patient 1 worked in the office building, in the tested room 1.

Ochratoxin and citrinin have been identified in raisins, coffee, cereals, wine and beer; enniatin and zearalenone have also been commonly detected in cereal product. The symptoms experienced by our occupants related to the workplace and their diets were unchanged. Therefore, the inhalation route of entry remains as a possible option. In our view, even minute amounts of mycotoxins inhaled from indoor air may be a potential health hazard, since the levels of inhaled mycotoxins are dependent on relative humidity (RH %) and temperature.

We performed the urinalysis twice in Patients 1 and 2 and detected changes in the levels of mycophenolic acid; in Patient 1, the concentration decreased, whereas in Patient 2, it increased after 2.5 months. Many mycotoxins are better extracted and dissolved in relatively non-polar solvents thus leading to the assumption that they are soluble in fatty tissues (reviewed in [2]). Therefore, their excretion kinetics would be expected to depend on an individual’s mass of body fat as well as their genetic background [55]. Unfortunately, we did not examine the home of Patient 2. We emphasize that it is important to compare the mycotoxin profiles of the occupants who have had similar exposures, since it is very likely that in persons exposed to the same ecological system, their mycotoxin profile will display similarities. As an example, Patient Y (unrelated to the working community) excreted high levels of gliotoxin, whereas Patients 1 and 2 excreted mycophenolic acid, and the levels remained high in the follow-up.

The strength of this study is the comprehensive and holistic multidisciplinary approach to assessing not only the patients but also the environments where they lived and worked. The environment was studied not only by accepted conventional microbiological techniques but also by a novel indoor air condensation technique supported by the functional cytotoxicity tests.

The importance of this communication is the idea of adopting a holistic approach to help the occupants to identify health risks of indoor air e.g., due to the presence of molds. The cumulative or prolonged exposure to toxic indoor air will inevitably be detrimental when the reversible SBS will be transformed into an irreversible DMHS with the so-called loss of tolerance to many unrelated compounds. This may lead to the development of multiple chemical sensitivity (MCS), chronic fatigue syndrome (CFS), autoimmune diseases, debilitating neurological functions, and disturbances in peripheral nervous system, to mention only a few hazardous outcomes.

The limitations of this study are as follows: Firstly, we have investigated a small working community and therefore this study is a pilot. Secondly, we were unable to explicitly demonstrate that the condensate of the workplace indoor air contained the same mycotoxins as the urine from Patients 1 and 2, but the findings were suggestive that this was the case. In the future, it would be valuable to apply our holistic approach as presented in Figure 3 as a roadmap to explain the morbidity caused by toxic indoor air. This approach might prove beneficial to encourage collaboration and cooperation between the treating physicians, environmental scientists, public health care providers and insurance companies.

## 5. Conclusions

We utilized a multidisciplinary, etiology-orientated approach for studying the morbidity associated with the toxic indoor air. We used (1) a technique for the collection of condensed indoor air water samples, (2) functional cytotoxicity tests to assess the toxicity of the indoor air condensate, (3) clinical evaluation of a working community exposed to poor indoor air, and (4) urinalysis for mycotoxin detection. We detected mycophenolic acid in the urine in both of the two studied occupants of the same moisture-damaged building. This potentially immunosuppressive mycotoxin can cause immune dysregulation that in the long run may be related to increased oncologic morbidity and susceptibility to infections.

Our recommendations are: 1. Examine the patients keeping in mind the possibility of a systemic and a multi-organ disease with high inter-individual variability; 2. Perform appropriate laboratory tests for toxicosis, depending on the clinical symptoms; 3. Undertake a urinalysis for mycotoxins; 4. Study the patient´s environment using microbiological and toxicological methods, especially techniques capable of detecting particulate matter, volatile organic compounds (VOC) of gases and indoor condensed water; 5. Recommend that patients should avoid the inhalation of toxic indoor air and provide prompt rehabilitation. If possible, one should attempt to identify the source of the health risk and in this respect, it is recommended that the toxicity of the indoor air should be investigated and a urinalysis also performed on their household contacts (HHC).

Larger studies are needed to test the validity of our holistic approach.

## Figures and Tables

**Figure 1 jof-08-00104-f001:**
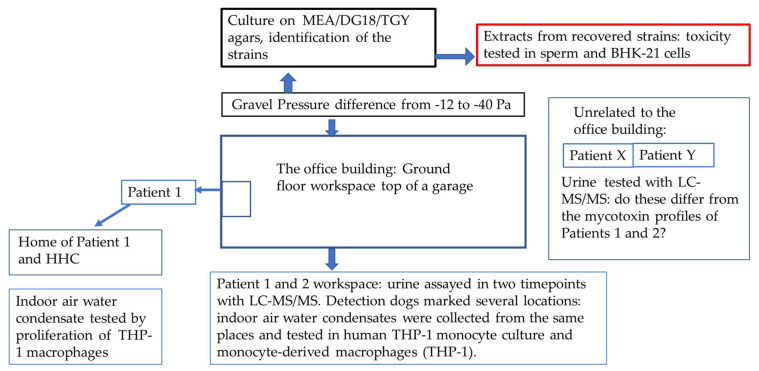
The study design.

**Figure 2 jof-08-00104-f002:**
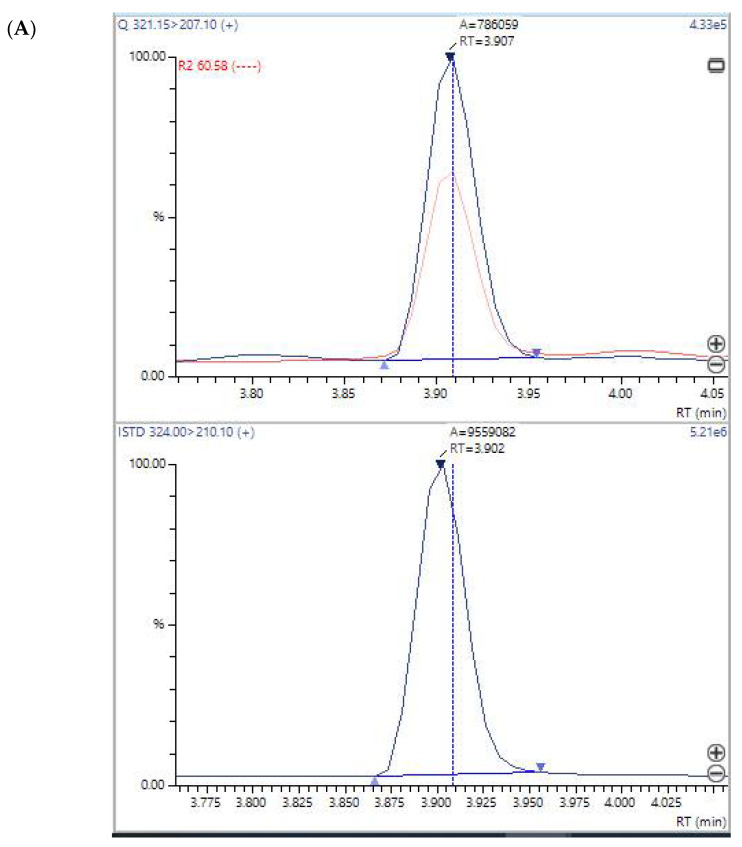
(**A**) LC-MS/MS plot of mycophenolic acid in the urine sample of Patient 1 above and the internal standard below. The graph shows a tracing in blue which is the quantifier ion (mass = 207.1) and a tracing in red which is the qualifier ion at mass = 159.1. The peak below with only a single blue tracing is the tracing of the deuterium-labeled mycophenolic internal standard (mass = 210.1). The mycophenolic acid internal standard has a virtually identical retention time to mycophenolic acid in the patient, but the quantifier ion is mass = 210.1 because of three deuterium atoms in the internal standard; (**B**) LC-MS/MS plot of mycophenolic acid in the urine sample of Patient 2. The upper graph of the patient shows a tracing in blue which is the quantifier ion (mass = 207.1) and a tracing in red which is the qualifier ion at mass = 159.1. The peak below with only a single blue tracing is the tracing of the deuterium-labeled mycophenolic internal standard (mass = 210.1).

**Figure 3 jof-08-00104-f003:**
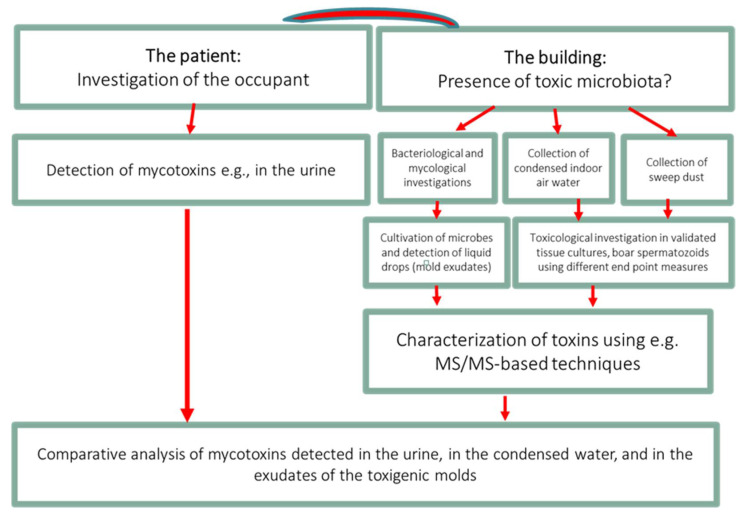
The roadmap to prove causality between the symptoms experienced by the occupant and the exposure to toxic indoor air.

**Table 1 jof-08-00104-t001:** Estimation of the inhaled water during the stay in a problematic building.

	Inhaled Air Vol (L)	Number of Inhalations	Inhaled m^3^ of Air/8 h	RH (%)	Temp (°C)	Humid Ratio	Inhaled Water/8 h/mL/g Water
Space 1 (Pat 1)	0.5	13	3.12	30.8	21.4	4.877	15.22
Space 2	0.5	13	3.12	30.2	21.4	4.781	14.92
Space 3	0.5	13	3.12	30.6	21.6	4.905	15.30
Space 4 (Pat 2)	0.5	13	3.12	29.1	21.6	4.663	14.55

**Table 2 jof-08-00104-t002:** The symptoms experienced by the working community. The symptoms often reported by patients with chronic mold illness are marked with *.

	Patient’s History	Symptoms	Findings during the Visit
No 1	No previous exposure to dampness microbiota	* Irritation of eyes	* Symptoms became worse when entering the problematic building
	No change in the diet	* Very severe headache	* Symptoms did not relieve during weekends
		*Recurrent sinusitis	Status uneventful
		* Multiple courses of antibiotics	The lesions were indurated, watery and had been scratched except for dermatitis on hands, legs, stomach, knees and back
		* Thermoregulation problems	
		* Itching of the skin	
		* Exanthema	
		* Sore throat	
		* Gastrointestinal reflux for months	
		* “Brain fog”	
		* A flu-like feeling	
No 2	Lived in a town house with his/her family	* Dyspnoea	* Symptoms had started to ease in 2 weeks if the person was absence from work
	All the other family members were asymptomatic	* Migraine	* Large psoriatic lesions especially on the elbows, back, scalp and chest, lips were cracked open
		* Phlegm	* The blood pressure was elevated
		* Muscle pain	
		* Fatigue	
		* Concentration problems	
		* Prolonged cough	
		* Sleeping problems	
No 3	Previously suffered from gastrointestinal problems and allergy	* Sore throat	*Redness of eye conjunctiva, unrelated to season
		* Blisters on the oral mucosa	Exanthema in the lower neck region
		* A flu-like feeling	
		* Blurred vision	
		* Severe fatigue	
		* Dyspnea	
		* Joint pain	
		Irregular peristaltic action and	
		menorrhagia	
No 4	Gastrointestinal dysbiosis, leaky gut	* Thermoregulation problems	Otherwise, the status uneventful
		* Two flu-like episodes	Axillary temperature 36.8 °C, exanthema in the lower neck region
		* Increased sputum production	
No 5	Previously healthy, the individual had not been exposed to dampness and mold	* Two courses of antibiotics for	Status uneventful
		tonsillitis	
		* Sore throat	
		* Evenings: eyes dry and voice hoarse	
		* Occasional instances of fatigue	
		* Memory problems	
No 6	Did not attend the doctor’s office	Was interviewed by telephone	

**Table 3 jof-08-00104-t003:** Detection of mycotoxins in urine by LC-MS/MS.

Mycotoxins Reference Values in ng/g Creatinine	Patient 1 First Test	Patient 1 Follow-Up Test	HHC to Patient 1 (Time-Point as for Pat 1 Follow-Up)	Patient 2 First Test	Patient 2 Follow-Up Test	Patient X	Patient Y
Aflatoxin M1 (3.5–20)	0	0	0	0	0	0	0
Ochratoxin A (4–20)	11.23	9.82	* 63.52	7.01	10.96	8.58	18.36
Gliotoxin (200–2000)	0	0	0	0	0	0	* 910.98
Sterigmatocystin (0.2–1.75)	1.14	0	0	0.74	0	0	0
Mycophenolic acid (5–50)	* 284.64	30.45	0	* 50	* 130.98	17.79	7.11
Roridin E (1–6)	0	0	0	0	0	0	0
Verrucarin A (1–10)	0	0	0	0	0	0	0
Enniantin B (0.07–1)	<0.07	0	0	0	0	0.43	0
Zearalenone (0.5–10)	0	0	0	4.36	* 15.54	5.55	0
Chaetoglobosin A (20–80)	0	0	0	0	0	0	0
Citrinin (10–50)	<10	<10	14.9	17.47	21.56	0	0

The * indicates excessively high values.

**Table 4 jof-08-00104-t004:** Testing of indoor air condensates collected from the office.

Sample	RH %	T °C	Change in Cell Viability %, Mean ± stdev, *p*	
			THP-1 Monocytes	THP-1 Macrophages
Working room 1	30.8	21.4	22.26 ± 20.81 *	−9.54 ± 1.25 ***
Sample 2	30.2	21.4	18.12 ± 16.6	−2.87 ± 2.06 *
Sample 3	30.6	21.6	7.66 ± 10.70	−6.26 ± 2.40 ***
Sample 4	29.1	21.7	28.40 ± 12.59 ***	−6.62 ± 1.26 ***

Patient 1 worked in room 1, sample 4 was taken close from working area of Patient 2. The toxicity of the condensates was tested at a 10% concentration on THP-1 monocytes and THP-1 macrophages. The results are normalized against control and expressed as % change in cell viability, mean ± stdev, as compared to the control. Negative values refer to a decrease in cellular viability, positive values refer to an increase in mitochondrial activity or proliferation. Both are considered as adverse effects. Each was tested in six parallel samples. The statistically significant changes in viability are bolded and indicated as * *p* < 0.05; and *** *p* < 0.001.

**Table 5 jof-08-00104-t005:** Testing of indoor air condensates collected from the home of Patient 1 and the HHC. Toxicity of condensates on THP-1 macrophage when tested at 10% and 25% concentrations. The methods and explanations are as in Table 4.

Sample	RH %	T °C	Change in THP-1 Macrophage Viability %, Mean ± Stdev, *p* (Significance)	
			10% Condensate	25% Condensate
Dormitory 1st floor (Patient 1 and HHC)	35.6	22.4	−2.13 ± 4.55	−13.35 ± 5.84 ***
Bedroom (son) 2nd floor	37.8	22.2	1.78 ± 3.02	1.56 ± 3.72
Bedroom (daughter) 2nd floor	38.7	22.0	6.33 ± 10.30	1.07 ± 8.56
Living room 2nd floor (HHC worked in this space)	39.6	22.0	2.83 ± 6.17	4.50 ± 4.65 *

The statistically significant changes in viability are bolded and indicated as * *p* < 0.05; and *** *p* < 0.001.

**Table 6 jof-08-00104-t006:** Toxic colonies: A total of 32 colonies were tested for toxicity to cultured cells.

Sample	Fungal Genus or Group	Culture Plate	Toxicity in Sperm Test	Toxicity in BHK-Test
Sample 2	*Penicillium*	MEA	Toxic	No
Sample 4	*Penicillium*	MEA	Toxic	No
Sample 9	*Acremonium* *sensu lato*	MEA	Toxic	Toxic
Sample 16	*Acremonium* *sensu lato*	MEA	Toxic	Toxic
Sample 18	*Penicillium*	DG18	Toxic	No
Sample 19	*Penicillium*	DG18	Toxic	No
Sample 21	*Aspergillus ochraceus-group*	DG18	Toxic	No
Sample 27	*Aspergillus ochraceus-group*	DG18	Toxic	No
Sample 28	*Aspergillus section* *Aspergillus (Eurotium)*	DG18, direct cultivation	Toxic	Toxic
Sample 29	*Aspergillus section* *Aspergillus (Eurotium)*	DG18, direct cultivation	Toxic	Toxic
Sample 32	Bacterium	TGY	Toxic	No

## Data Availability

Not applicable.

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
