# Peer review of "Toxic Indoor Air Is a Potential Risk of Causing Immuno Suppression and Morbidity—A Pilot Study"

_jof, 2022, doi:10.3390/jof8020104_

Round 1

Reviewer 1 Report

The aim of the investigation was to establish a connection between exposure to mold and certain diseases. In many respects, however, it does not meet the criteria of a scientific investigation, so that I would urgently advise against publication.

On the one hand, the number of people examined was far too small. On the other hand, the investigations were carried out very inconsistently. The selection of the examined persons is also unclear, and above all their biological exposure was not examined in detail, so that it is not possible to establish a relation of corresponding exposures to diseases allegedly present in them.

There were only 6 study participants in total, one of whom dropped out before the start of the study. They should be examined to see whether they have become ill from staying in their office building and the alleged mold exposure there. Two other people unrelated to the office building were also included in the study. A household contact of one of the office staff was also included.

Obviously, certain symptoms were recorded in the remaining 5 office workers, although it remains unclear how this happened. It is likely that these symptoms were only captured from people's reports. It remains open whether this was done in the form of standardized interviews or whether the people were asked explicitly about the individual symptoms. In general, many different symptoms, which were often very unspecific, were recorded in the 5 people. It is generally not possible to assign these different and unspecific symptoms to a common cause.

The control persons were apparently not asked about their symptoms. Since the people examined supposedly wanted to remain anonymous, there is no further information on them, e.g. regarding age, gender and previous illnesses.

Exposure to mold or other biological agents, which are presented to be the only cause of the reported symptoms, was not recorded, either in the offices or in the private living quarters of the persons examined. Obviously, no other possible causes of the reported symptoms were searched for either.

Instead of carrying out systematic mold measurements in the room air, the offices were only inspected with the help of a tracking dog. In addition, apparently only a single material sample from a cavity adjacent to the office, which is located behind a plasterboard, was examined for mold. The mold exposure is completely unknown in the two control persons and in the private household, which was also considered for one of the persons working in the office.

Obviously, the current nomenclature was not used for the sample examined for mold. The statement "possible Streptomycetes were identified" also indicates uncertainties in speciation.

Part of the study consisted of testing urine samples for mycotoxins. However, these studies were only carried out for two study participants and three control persons, whereby contact from the private household was used as a further control person in addition to the two other control persons.

The fact that higher mycophenolic acid concentrations were found in the two people working in the office was presented as the most important result of the study. However, mycotoxins were also found in the urine of the household contact person and a control person. In addition, there was a decrease of the mycophenolic acid concentration in one of the two people working in the office 2.5 months later after the working community had moved to a clean building and an increase in in the urine of the other person of the office. The other mycotoxins measured in the urine of these two people did not give a consistent picture either.

In addition, hardly anything can be read in the LC-MS / MS chromatograms on which these measurement results are based. It is also not clear which samples are shown exactly in the chromatograms. The chromatograms for the control persons are also missing.

The examined condensate samples, which were used for toxicity measurements, were also not examined for their biological composition. They were supposed to be examined for mycotoxins, but this failed because they were not stored and sent to the laboratory for mycotoxin analysis in an adequate manner. This  was then cited as a possible explanation for the fact that no mycotoxins were found in these samples, although there was much argument about the toxicity of these condensates throughout the study.

While the condensate samples from the office building were examined with two toxicity assays, the additionally examined condensate samples from the household of one of the two office person were examined with only one assay. It remains open why no condensate samples were examined in the environment of the control persons.

The condensate sample from the household also showed toxic effects in these tests, but only at slightly higher concentrations. The number of samples for the toxicity tests was far too low to make clear statements here, and moreover the results were very inconsistent. The results of the controls are not shown at all. It is also unclear in these tests why both positive and negative deviations in cell viability are considered as adverse.

Toxicity tests were also made for the fungal colonies that grew after culturing the material sample from the hollow area in the proximity of the floor of the office. However, different tests were used here and for the condensates. Here, too, the results were inconsistent for the two tests used. It is also not surprising that such high concentrations, as used here for the cell exposure to the fungal colonies, lead to effects on the cells. Unfortunately, no mycotoxin analysis was carried out for the fungi detected in this sample.

In general, the description of the methods (e.g. for LC-MS/MS analysis) is inadequate. The examined rooms are also not described in detail. It remains unclear where the various measuring points were located and how they are related to the whereabouts of the people. In general, it is difficult to understand what exactly was done with which samples and how the entire study design looked like.

In the discussion, the slightly higher mycophenolic acid detection in some of the urine samples is used to speculate on possible cancer diseases caused by mold growth, which is completely nonserious.

Since hardly any relations were found in this “pilot study”, there is no reason to expand this research concept, which is suggested by the authors.

For several abbreviations, e.g. TT, DM or HHC, there is no explanation of what they stand for.

Under Table 1, the respiratory rate is incorrectly stated as 13 times/h.

Author Response

Point-by point explanation to the changes done to the manuscript.

First, we want to thank all reviewers for their valuable time and  reading our manuscript.

We did our best to describe a new approach when the occupants with indoor air problems present to the medical doctor. Indeed, their symptoms are usually non-specific and high interindividual variation is observed. There is also no single laboratory method that may help the clinician to do diagnosis. The use of urinalysis to detect mycotoxins is not common in medical practice. The use of condensed water sample for toxicological analysis is novel. Using only microbiological work-up in a problematic building will not always lead to the understanding of causality. We claim that our study is a pilot, and we advocate that this approach should be repeated either to confirm or to refute our view of a holistic investigation. The Referee 1 is welcome to do reproduce our study and to publish his/her results and to criticize us publicly if our approach does not prove useful.

For smaller notes see the attached file.

Reviewer 2 Report

The aim is stated clear. The authors stated clearly what study found and how they did it. The title is informative and relevant.

The references are relevant and recent. Appropriate and key studies are included.

The research question also justified given what is already known about the topic. The process of selection of the subjects was clear. The study methods are valid and reliable. There are enough details provided in order to replicate the study.

The data is presented in an appropriate way. Results are discussed from different angles and placed into context without being overinterpreted.

The conclusions answer the aim of the study. The conclusions are supported by references and own results.

The limitations of the study are not fatal, but they are opportunities to inform future research.

Specific comments on weaknesses of the article and what could be improved:

Major points  

  1. Please, extend the conclusion by emphasizing in detail what are the probable negative outcomes of the workers related to the toxic indoor air. 

Minor points

  1. Please, remove the full stop after the titles endings
  2. Could you please discuss the clinical implications of the results - what would be your recommendations?

Author Response

Point-by point explanation to the changes done to the manuscript.

First, we want to thank all reviewers for their valuable time and  reading our manuscript.

According to the reviewer 2, we suggest a comment to the end, from line 456:

The importance of this communication is the idea of a holistic approach in helping the occupants to identify health risks of indoor air due to e.g. presence of molds. The cumulative or prolonged exposure to toxic indoor air will inevitably lead to the chronic cause of the disease when the reversible SBS will turn to irreversible DMHS with the so-called loss of tolerance to many unrelated compounds. This may lead to the development of multiple chemical sensitivity (MCS), chronic fatigue syndrome (CFS), autoimmune diseases, debilitating neurological functions, and dysautonomy of peripheral nervous system, to mention a few hazardous outcomes.

Also in the conclusion (starting from line 485) we have added as suggested by reviewer 2:

This potentially immunosuppressive mycotoxin can cause immune dysregulation that in the long run may be related to increased oncologic morbidity and susceptibility to infections.

Our recommendations are: 1. Examine the patients keeping in mind a systemic and a multi-organ disease with high inter-individual variability; 2. Perform appropriate laboratory tests of toxicosis, depending on clinical symptoms; 3. Execute urinalysis for mycotoxins; 4. Study using microbiological and toxicological methods the patient´s environment, especially for particulate matter, volatile organic compounds (VOC) of gases and indoor condensed water; 5. Recommend patients to avoid inhalation of toxic indoor air and provide prompt rehabilitation. The help to identify the source of health risk it is recommended to investigate the indoor air toxicity and to perform urinalysis of the household contacts (HHC), too. 

Reviewer 3 Report

The authors present a pilot study that attempts to establish a relation between the symptoms of people that have occupied a toxic damp building for a long time and etiologic agents. The authors use multidisciplinary approach combining clinical history, microbiology, clinical toxicology, cytotoxicity, to try and establish the etiologic relationship etc. The environment was studied using water condensates to determine air toxicity. 

The study has limitation which have been addressed by the authors and potential suggestion made with reasonable conclusions and future recommendations based on the study.

1. A number of spelling mistake present must be addressed:

e.g line 51-please check the spelling of 'accociatewd'

line 339 and 340, please check spelling  'condenced'  

Please check  the spelling 'increaded' in 459

Comment and suggestions

2. Please provide a lend to explain the different colors of the LC graphs in figure 2

 3. I suggest to make table 2 more easy readable the authors may consider columns under titles such as  Patient, background, significant symptoms and status - to make table much more easier to read 

Author Response

First, we want to thank all reviewers for their valuable time and  reading our manuscript.

  1. A number of spelling mistake present must be addressed:

e.g line 51-please check the spelling of 'accociatewd' corrected associated

line 339 and 340, please check spelling  'condenced'    corrected condensed

Please check  the spelling 'increaded' in 459  corrected increased

Comment and suggestions

  1. Please provide a lend to explain the different colors of the LC graphs in figure 2. The colors will be explained when we will get answer from the Great Plains laboratory.
  2. I suggest to make table 2 more easy readable the authors may consider columns under titles such as  Patient, background, significant symptoms and status - to make table much more easier to read. The table is changed, and it is indeed better, but needs help from the journal to look nicer.